# Proto-SaGa: Prototype-based 3D Scene Segmentation with Semantic-aware Gaussian Grouping

## Abstract

Segment anything models (SAM), trained with lots of ground-truth labels, have achieved strong performance in 2D scene segmentation. Compared to this, accurate 3D scene segmentation remains challenging, since annotating consistent segmentation masks across multiple views is highly labor-intensive. To address this, many approaches have been proposed using inconsistent masks predicted by SAM as pseudo labels. They typically build on 3D Gaussian splatting (3DGS) to synthesize and segment novel views in a 3D scene simultaneously. To be specific, several 3DGS-based methods focus on associating the inconsistent masks across training views so that a classifier is trained with the associated masks. They however have two limitations: (1) the association process considers only the location of each 3D Gaussian in the scene and (2) training a classifier with the associated masks is prone to overfitting to incorrect labels of the associated masks. We introduce in this paper Proto-SaGa, a novel 3DGS-based framework that addresses the aforementioned limitations. Specifically, we present a semantic-aware mask association strategy that exploits both location and high-level semantics of each Gaussian to improve the consistency of the associated masks. We also propose a novel inference scheme that alleviates the influence of possibly incorrect results within the associated masks. Specifically, we obtain a set of prototypes by averaging features with the consistent masks, and use it as a classifier at test time without further training. Extensive experiments on Replica, LERF-Mask, ScanNet, and Mip-NeRF 360 demonstrate the effectiveness of our approach. We will make our code publicly available upon acceptance.

## 1 Introduction

Recent approaches to localizing objects in 2D images have achieved remarkable progress thanks to large-scale datasets. For instance, segment anything models (SAM) (Kirillov et al., 2023) leverage 11M images along with 1B high-quality masks at training time, producing notable improvements in 2D scene understanding. Compared to this, 3D scene segmentation (Dai et al., 2017; Schult et al., 2023) has shown limited advancements, since annotating consistent segmentation masks in a 3D scene is extremely labor-intensive.

To alleviate the annotation cost, many approaches have been introduced exploiting 2D foundational models (Kirillov et al., 2023; Caron et al., 2021; Radford et al., 2021) for 3D scene segmentation. They typically build on novel view synthesis methods (Mildenhall et al., 2020; Kerbl et al., 2023), synthesizing novel views and producing consistent segmentation masks simultaneously. Early work (Kobayashi et al., 2022) adopts neural radiance fields (NeRF) (Mildenhall et al., 2020) and trains additional feature fields that imitate feature representations extracted from CLIP (Radford et al., 2021) and DINO (Caron et al., 2021). Although NeRF-based methods (Cen et al., 2023; Siddiqui et al., 2023) provide decent segmentation results in novel views, they are limited in that the volumetric rendering of NeRF is computationally expensive and time-consuming. For faster rendering, several approaches rely on 3D Gaussian splatting (3DGS) (Kerbl et al., 2023), and augment each 3D Gaussian with an additional embedding vector that is used to render a feature map for segmentation at a specific view. 3DGS-based methods typically exploit segmentation masks predicted by SAM to learn discriminative features, but the segmentation masks are inconsistent across

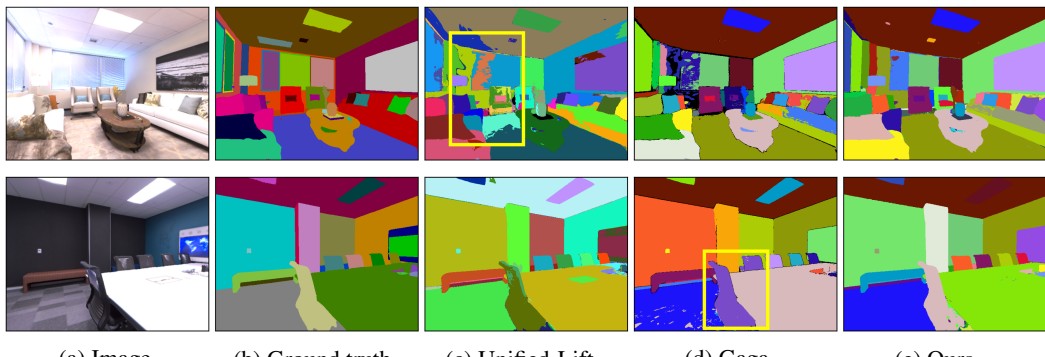

(a) Image.  (b) Ground truth.  (c) Unified-Lift.  (d) Gaga.  (e) Ours.

Figure 1: We compare segmentation results of Unified-Lift (Zhu et al., 2025), Gaga (Lyu et al., 2024), and our method on Replica (Straub et al., 2019). Yellow boxes highlight incorrect predictions.

different views, making it difficult to produce consistent segmentation results. To address this, recent approaches (Zhu et al., 2025; Ying et al., 2024) propose to leverage a contrastive clustering strategy (Li et al., 2020), enabling learning discriminative embeddings. They are however sensitive to the number of clusters, producing imprecise segmentation results, especially for wall and floor (Fig. 1(c)). Another line of work (Lyu et al., 2024; Ye et al., 2024) instead introduces a mask association scheme that generates pseudo segmentation masks, which are consistent across views, either by using an off-the-shelf video object tracker (Cheng et al., 2023) or by grouping 3D Gaussians directly. The associated masks are then used as pseudo labels to train a classifier that takes rendered feature maps as input. While the mask association is effective, the quality of pseudo labels largely influences the segmentation performance. For example, if two different chairs are incorrectly grouped as one within the pseudo-labels, the segmentation results also reflect the same error (Fig. 1(d)).

We introduce in this paper a novel framework for 3DGS-based segmentation, dubbed Proto-SaGa, that generates consistent segmentation masks across training views and use them to obtain a prototype-based classifier for inference. Specifically, we present a semantic-aware mask association strategy that better groups 3D Gaussians, where each group represents an individual object in a 3D scene. To this end, we define a view-specific classifier for each training view, and train these classifiers using inconsistent masks obtained from SAM as pseudo labels. After training, we associate each region within the inconsistent masks at every training view with a set of 3D Gaussians based on two criteria: (1) the distance of each Gaussian from the 2D image plane and (2) the softmax probabilities of each Gaussian computed by the learned classifier. Different from Gaga (Lyu et al., 2024) that uses the first criterion only, our approach incorporates high-level semantics (*i.e.*, the second criterion), better associating the inconsistent masks across views. After the association process, we discard the view-specific classifiers. While a straightforward way to using the associated masks is training a new classifier shared across different views as in current methods (Lyu et al., 2024; Ye et al., 2024), it could be prone to association errors. To address this, we introduce a prototype-based segmentation pipeline that exploits the associated masks to obtain a prototype-based classifier, rather than using them to train a new classifier. In particular, we first render a feature map at each training view, and compute a set of prototypes by averaging features belonging to the same region within the corresponding mask. We then average the prototypes across training views to obtain an ensemble of prototypes, which is used as a classifier to render coherent segmentation masks in novel views. Our approach to using the prototype-based classifier mitigates the influence of potentially incorrect results from the association process. We evaluate our approach on standard benchmarks (Straub et al., 2019; Ye et al., 2024) to demonstrate its effectiveness. Experimental results show that our approach outperforms current methods by a significant margin, providing precise and coherent segmentation results (Fig. 1(e)). Our main contributions are summarized as follows:

- We propose a semantic-aware mask association strategy that uses location and high-level semantics of each Gaussian, producing coherent masks across training views.

- We introduce a novel inference scheme using a prototype-based classifier, allowing us to alleviate erroneous results that possibly occur during the association process.

- We present comprehensive experiments on standard benchmarks and show that our approach achieves a new state of the art.

## 2 RELATED WORK

**Novel view synthesis.** There have been approaches (Mescheder et al., 2019; Park et al., 2019; Sitzmann et al., 2019) that adopt implicit scene representations to avoid the discretization error of explicit counterparts (*e.g.*, voxel girds), where they typically train neural networks to represent a 3D scene. Among them, NeRF (Mildenhall et al., 2020) proposes to exploit a multi-layer perceptron (MLP) that maps a 3D point into its color and volume density in a specific viewing direction. Although NeRF achieves impressive results in novel view synthesis using a set of posed images at training time, the volumetric rendering process of NeRF requires forwarding a large set of 3D points through the MLP, which is computationally demanding. 3DGS (Kerbl et al., 2023) has recently been introduced as an effective alternative, representing a 3D scene with a set of 3D Gaussians explicitly. It synthesizes a novel view by projecting 3D Gaussians onto the corresponding image plane and applying an alpha-blending technique in a depth-sorted order. This allows to achieve real-time rendering, while producing high-quality results in novel views. Moreover, compared to NeRF, the explicit nature of 3DGS enables manipulating and editing of 3D scenes more user-friendly, which makes it practical for interactive 3D scene segmentation. Based on these benefits, we build our approach on top of 3DGS to synthesize novel views and generate coherent segmentation masks.

**3D scene segmentation.** With the recent success of novel view synthesis, many approaches for 3D scene segmentation have been proposed. Many methods rely on NeRF (Mildenhall et al., 2020) or 3DGS (Kerbl et al., 2023) to synthesize and segment novel views simultaneously. To be specific, NeRF-based methods (Cen et al., 2023; Siddiqui et al., 2023) introduce an additional feature field to imitate features extracted from CLIP (Radford et al., 2021), enabling open-vocabulary recognition in a 3D scene. They however entail the volumetric rendering process, which is time-consuming, limiting the applicability in real-world scenarios (*e.g.*, interactive editing). On the contrary, benefiting from the real-time rendering ability, 3DGS-based methods have proven effective in 3D scene segmentation. Specifically, several approaches (Qin et al., 2024; Li et al., 2025; Zhou et al., 2024) attempt to distill rich semantics from 2D foundational models (Radford et al., 2021; Kirillov et al., 2023; Li et al., 2022) into 3D Gaussians at training time. For example, LangSplat (Qin et al., 2024) and Feature 3DGS (Zhou et al., 2024) propose to imitate features extracted from CLIP and LSeg (Li et al., 2022), respectively. Although these methods are effective in localizing objects of a certain class (*i.e.*, semantic segmentation), they struggle to distinguish individual objects of the same class (*i.e.*, instance segmentation). Rather than imitating features from the 2D foundational models, recent approaches (Cen et al., 2025; Ying et al., 2024; Zhu et al., 2025) focus on learning discriminative features to address both semantic and instance segmentation. They first apply SAM (Kirillov et al., 2023) to each training view independently to obtain segmentation masks, and then adopt a contrastive learning framework (Li et al., 2020) that encourages features to be similar if they belong to the same region within a segmentation mask at a specific view. Since the segmentation masks predicted by SAM are inconsistent across different views, inferring segmentation results at novel views requires grouping features with a clustering technique (*e.g.*, HDBSCAN (McInnes et al., 2017)). The clustering scheme is however sensitive to the number of clusters, leading to suboptimal segmentation results. Instead of grouping features, Gau-Grouping (Ye et al., 2024) employs off-the-shelf video object trackers (Cheng et al., 2023) to associate inconsistent masks. The associated masks are then used as pseudo ground-truth labels to train additional embeddings attached to each 3D Gaussian. The video object tracker however suffers from handling significant changes between training views, producing inaccurate results. To address this, Gaga (Lyu et al., 2024) first trains vanilla 3DGS and then groups 3D Gaussians directly. In particular, it identifies a set of 3D Gaussians belonging to each region within the inconsistent mask at every training view, and determines whether each pair of regions from two different views represents the same object in a given 3D scene based on the number of overlapping Gaussians. However, Gaga uses only the depth of each Gaussian to associate each region with 3D Gaussians, which often leads to unsatisfactory results. Our approach differs in that we take account both depth and semantics of each Gaussian for better association.

## 3 METHOD

In this section, we provide a detailed description of our approach. Specifically, we introduce a simple yet effective method for training a set of 3D Gaussians along with a separate classifier at each training view (Sec. 3.1). We then present a semantic-aware grouping strategy that clusters the

Figure 2: An overview of our training process. At each training step, we randomly select a training view and synthesize its image and feature map. We adopt the same objective as in 3DGS (Kerbl et al., 2023) to reconstruct the given 3D scene, while using a cross-entropy loss to supervise the feature map for segmentation. Specifically, we define an individual classifier at each training view, and train these classifiers with the inconsistent mask predicted by SAM (Kirillov et al., 2023).

3D Gaussians to obtain consistent segmentation masks (Sec. 3.2), and describe a novel inference pipeline using prototypes (Sec. 3.3).

## 3.1 TRAINING

Following the common practice (Ye et al., 2024; Lyu et al., 2024; Zhu et al., 2025), we build our method on 3DGS (Kerbl et al., 2023) to synthesize and segment novel views simultaneously (Fig. 2). To be specific, we augment each 3D Gaussian with an additional embedding vector, and define the $i$-th 3D Gaussian, $G_i$, as follows:

$$G_i = \{p_i, s_i, q_i, \alpha_i, c_i, f_i\}, \tag{1}$$

where $p_i$, $s_i$, $q_i$, $\alpha_i$, and $c_i$ indicate its center, scale, orientation, opacity, and spherical harmonics (SH) coefficients, respectively. We denote by $f_i$ the $D$-dimensional embedding of the $i$-th Gaussian. We can render a color value at pixel $\mathbf{p}$ from the $k$-th view, $\hat{C}_k(\mathbf{p})$, by projecting 3D Gaussians onto the corresponding 2D image plane as follows:

$$\hat{C}_k(\mathbf{p}) = \sum_{i \in N} c_i \alpha_i \prod_{j=1}^{i-1} (1 - \alpha_j), \tag{2}$$

where $N$ is the number of Gaussians ordered by the distance from the image plane (*i.e.*, depth). To supervise the rendered image, we adopt the same objective used in 3DGS with a balance parameter $\lambda$ as follows:

$$\mathcal{L}_{\text{REC}} = (1 - \lambda) \sum_{\mathbf{p}} \|\hat{C}_k(\mathbf{p}) - C_k(\mathbf{p})\|_1 + \lambda \mathcal{L}_{\text{SSIM}}, \tag{3}$$

where $C_k(\mathbf{p})$ is a ground-truth color and $\mathcal{L}_{\text{SSIM}}$ indicates a SSIM (Wang et al., 2004) loss, defined as follows:

$$\mathcal{L}_{\text{SSIM}} = 1 - \sum_{\mathbf{p}} \text{SSIM}(\hat{C}_k(\mathbf{p}), C_k(\mathbf{p})). \tag{4}$$

Similar to Eq. 2, we can also render a feature for segmentation at pixel $\mathbf{p}$ from the $k$-th view, $\hat{F}_k(\mathbf{p})$, as follows:

$$\hat{F}_k(\mathbf{p}) = \sum_{i \in N} f_i \alpha_i \prod_{j=1}^{i-1} (1 - \alpha_j). \tag{5}$$

To supervise the rendered features, we exploit segmentation masks predicted by SAM (Kirillov et al., 2023). These masks are inconsistent across training views, since we apply SAM to each view independently. That is, each mask has a different number of instance labels, and the instance labels are not associated across views. To address this, we propose a simple yet effective method that assigns a separate classifier to each training view. Let us suppose we have the inconsistent mask

at the $k$-th training view, denoted by $M_k$, and it contains $L_k$ instance labels. We then refer to the view-specific classifier of size $D \times L_k$ for the $k$-th view as $w_k$, and use it to compute a softmax probability at pixel $\mathbf{p}$ from the $k$-th view, $\sigma_k(\mathbf{p})$, as follows:

$$\sigma_k(\mathbf{p}) = \text{Softmax}(\tau \frac{w_k^\top \hat{F}_k(\mathbf{p})}{|w_k||\hat{F}_k(\mathbf{p})|}), \tag{6}$$

where $\tau$ indicates a temperature parameter adjusting the sharpness of the probabilities. To optimize both the features and the classifier, we adopt a cross-entropy loss as follows:

$$\mathcal{L}_{\text{CE}} = \sum_{\mathbf{p}} \text{CE}\left(\sigma_k(\mathbf{p}), M_k(\mathbf{p})\right). \tag{7}$$

Since the feature map $\hat{F}_k$ is rendered from the features of individual Gaussians (*i.e.*, $f_i$), which are shared across all views, the cross-entropy term can guide Gaussians to learn discriminative features, even with view-specific classifiers. The overall objective for a given scene is then defined as follows:

$$\mathcal{L} = \mathcal{L}_{\text{REC}} + \lambda_{\text{CE}}\mathcal{L}_{\text{CE}}, \tag{8}$$

where we denote by $\lambda_{\text{CE}}$ a balance parameter. Unlike Gaga (Lyu et al., 2024), our approach learns to reconstruct the 3D scene and acquire discriminative features simultaneously.

## 3.2 ASSOCIATION

Following Gaga (Lyu et al., 2024), we group 3D Gaussians to obtain consistent segmentation masks across training views. Specifically, we first identify a set of Gaussians belonging to the $t$-th instance of the inconsistent mask $M_k$ at the $k$-th training view as follows:

$$\mathcal{G}_k(t) = \{G_i \mid \mathbf{u}_i \in \mathcal{R}_k(t) \text{ and } i = 1, 2, \ldots, E\}, \tag{9}$$

where $\mathbf{u}_i$ is the projected center of the $i$-th Gaussian on the image plane and $\mathcal{R}_k(t)$ indicates a set of pixels labeled as the $t$-th instance within $M_k$. Let us denote by $E$ the total number of elements in the set. Since the influence of Gaussians far from the image plane is negligible, Gaga proposes to filter them out as follows:

$$\mathcal{G}_k^{\text{d}}(t) = \{G_i \mid G_i \in \mathcal{G}_k(t) \text{ and } \text{Rank}_d[z_i] < \delta_d E\}, \tag{10}$$

where $z_i$ indicates the depth of the $i$-th Gaussian and $\delta_d$ is a hyperparameter for controlling the degree of filtering. $\text{Rank}_d[\cdot]$ is a function that returns the rank of the input among $E$ elements. Specifically, it assigns higher ranks to smaller inputs, that is, $\text{Rank}_d[z_i] < \text{Rank}_d[z_j]$ if $z_i < z_j$. Next, Gaga initializes a memory bank, denoted by $\mathcal{M}$, by using a set of Gaussians at the first training view (*i.e.*, $k$=1) as follows:

$$\mathcal{M}(t) = \mathcal{G}_1^{\text{d}}(t). \tag{11}$$

The memory bank is then updated repeatedly across the subsequent views. To be specific, Gaga defines an overlapping score between the $t$-th instance at the $k$-th training view ($k \neq 1$) and the $e$-th instance within the memory bank as follows:

$$\gamma_k(t, e) = \frac{\#[\mathcal{G}_k^d(t) \cap \mathcal{M}(e)]}{\#[\mathcal{G}_k^d(t)]}, \tag{12}$$

where $\#[\cdot]$ indicates a function counting the number of Gaussians. If the score is above the predefined threshold $\delta$, the corresponding set for the $e$-th instance within the memory bank is updated as follows:

$$\mathcal{M}(e) \leftarrow \mathcal{G}_k^d(t) \cup \mathcal{M}(e). \tag{13}$$

Otherwise, the set of Gaussians for the $t$-th instance at the $k$-th view, $\mathcal{G}_k^{\text{d}}(t)$, is appended to the memory bank as a new instance. After updating the memory bank across training views, the memory bank represents a unified set of instance labels across training views. However, simply using the distance from the image plane of each Gaussian as in Eq. 10 is suboptimal in that Gaussians near the plane do not always represent the corresponding instance.

**Semantic-aware memory bank.** To consider semantics of each Gaussian, we propose to use the learned classifiers. Formally, we compute the softmax probability of the $i$-th Gaussian for the $t$-th instance at the $k$-th training view as follows:

$$\sigma_{i,k} = \text{Softmax}(\tau \frac{w_k^\top f_i}{|w_k||f_i|}) \in \mathbb{R}^{L_k}. \tag{14}$$

Based on this, we define a new set of Gaussians for the $t$-th instance at the $k$-th view as follows:

$$\mathcal{G}_k^{\text{s}}(t) = \{G_i \mid G_i \in \mathcal{G}_k(t) \text{ and } \text{Rank}_s[\sigma_{i,k}(t)] < \delta_s E\}, \tag{15}$$

where $\sigma_{i,k}(t)$ is the probability value for the $t$-th instance and $\delta_s$ indicates a hyperparameter for eliminating Gaussians whose probability is low. $\text{Rank}_s[\cdot]$ is a ranking function that assigns higher ranks to higher inputs. Namely, $\text{Rank}_d[\sigma_{i,k}(t)] < \text{Rank}_d[\sigma_{j,k}(t)]$ if $\sigma_{i,k}(t) > \sigma_{j,k}(t)$. We propose to combine both criteria (*i.e.*, Eqs. 10 and 15) to identify a set of Gaussians belonging to the $t$-th instance at the $k$-th view as follows:

$$\mathcal{G}_k^{\text{saga}}(t) = \mathcal{G}_k^{\text{d}}(t) \cup \mathcal{G}_k^{\text{s}}(t). \tag{16}$$

The combined set, $\mathcal{G}_k^{\text{saga}}(t)$, is then used to construct and update the memory bank as in Eqs. 11, 12, and 13, better grouping 3D Gaussians. This allows us to build the memory bank that reflects how likely each Gaussian is to represent the corresponding instance.

### 3.3 INFERENCE

A straightforward way to predict segmentation masks at novel (*i.e.*, test) views is to group rendered features using HDBSCAN (McInnes et al., 2017) as in (Ying et al., 2024). However, it is susceptible to the number of clusters, leading to inaccurate segmentation results. Alternatively, similar to Gaga (Lyu et al., 2024), we could define a unified classifier and train it together with the features by using the consistent masks as pseudo labels. The additional training process however increases the computational cost, and is likely to overfit to the pseudo labels.

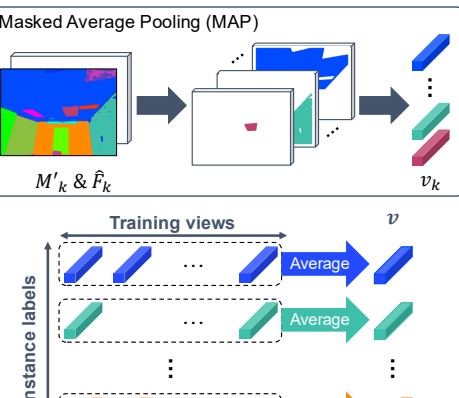

**Prototype-based segmentation.** We instead compute a set of prototypes at each training view, and average them across views to obtain a unified classifier without further training (Fig. 3). Concretely, we define the prototype for the $t$-th instance at the $k$-th training view, $v_k(t)$, as follows:

$$v_k(t) = \frac{1}{|\mathcal{R}'_k(t)|} \sum_{\mathbf{p} \in \mathcal{R}'_k(t)} \hat{F}_k(\mathbf{p}), \tag{17}$$

Figure 3: An illustration of our inference scheme. At each training view, we render a feature map and adopt masked average pooling to compute prototypes (top). We then obtain an ensemble of prototypes by averaging the prototypes for each instance label across training views (bottom).

where $\mathcal{R}'_k(t)$ is a set of pixels labeled as the $t$-th instance within the corresponding consistent mask. We then use an ensemble of prototypes to obtain a classifier weight for the $t$-th instance as follows:

$$v(t) = \frac{1}{K} \sum_k v_k(t), \tag{18}$$

where $K$ is the number of training views that contain the $t$-th instance. For inference, we predict an instance label at pixel $\mathbf{p}$ from the $n$-th novel view, $\hat{y}_n(\mathbf{p})$, as follows:

$$\hat{y}_n(\mathbf{p}) = \arg\max_t \left( \tau \frac{v(t)^\top \hat{F}_n(\mathbf{p})}{|v(t)||\hat{F}_n(\mathbf{p})|} \right), \tag{19}$$

where $\hat{F}_n$ is the rendered feature map at the $n$-th novel view. This allows us to segment novel views without using the clustering scheme, while preventing the overfitting problem.

Table 1: Quantitative comparison with state-of-the-art methods (Lyu et al., 2024; Ye et al., 2024; Zhu et al., 2025) on Replica (Straub et al., 2019). Numbers in bold indicate the best performance, while underscored ones represent the second best. I, P, and R indicate mIoU, precision, and recall, respectively. † indicates Unified-Lift (Zhu et al., 2025) trained with associated masks from the video object tracker (Cheng et al., 2023).

| Method | Metric | office_0 | office_1 | office_2 | office_3 | office_4 | room_0 | room_1 | room_2 | Avg. |
|---|---|---|---|---|---|---|---|---|---|---|
| Gau-Group | PSNR | 43.369 | 42.091 | 38.767 | 38.519 | 34.443 | 37.168 | 38.196 | 37.946 | 38.812 |
| | I | 19.7 | 36.0 | 20.6 | 16.3 | 19.5 | 20.1 | 27.8 | 17.0 | 22.1 |
| | P | 14.9 | 37.5 | 25.5 | 23.5 | 16.7 | 33.9 | 24.1 | 23.7 | 25.0 |
| | R | 16.7 | 40.9 | 17.4 | 14.6 | 14.3 | 20.4 | 24.1 | 14.5 | 20.4 |
| Gaga | PSNR | 44.496 | 43.330 | 39.582 | 39.370 | 36.035 | 38.150 | 39.612 | 38.753 | 39.916 |
| | I | 39.8 | 48.1 | 51.4 | 41.6 | 43.8 | 42.4 | 50.6 | 53.9 | 46.4 |
| | P | 22.1 | 41.2 | 49.2 | 41.7 | 43.7 | 41.7 | 46.0 | 57.4 | 42.9 |
| | R | 43.9 | 54.5 | 56.5 | 45.1 | 48.8 | 43.4 | 59.3 | 62.9 | 51.8 |
| Unified-Lift | PSNR | 44.548 | 43.104 | 39.665 | 39.437 | 36.050 | 38.210 | 39.655 | 38.828 | **39.937** |
| | I | 38.3 | 51.3 | 46.1 | 46.1 | 52.6 | 44.5 | 57.4 | 40.3 | 47.1 |
| | P | 15.4 | 24.2 | 36.5 | 39.4 | 29.1 | 35.5 | 32.6 | 37.7 | 31.3 |
| | R | 38.9 | 53.0 | 49.8 | 50.4 | 57.1 | 45.9 | 58.6 | 43.6 | 49.7 |
| Unified-Lift† | PSNR | 44.552 | 43.052 | 39.626 | 39.452 | 36.050 | 38.167 | 39.652 | 38.853 | 39.926 |
| | I | 25.0 | 44.9 | 31.9 | 24.7 | 31.3 | 22.9 | 40.6 | 21.5 | 30.4 |
| | P | 20.8 | 37.6 | 38.9 | 37.8 | 28.5 | 38.8 | 37.1 | 27.6 | 32.1 |
| | R | 23.9 | 49.2 | 30.4 | 25.6 | 31.0 | 20.8 | 46.3 | 19.9 | 30.9 |
| Ours | PSNR | 44.457 | 43.318 | 39.529 | 39.411 | 35.975 | 38.100 | 39.520 | 38.680 | 39.874 |
| | I | 44.6 | 52.1 | 55.7 | 46.7 | 48.8 | 50.1 | 58.9 | 56.5 | **51.7** |
| | P | 21.4 | 43.6 | 57.3 | 49.6 | 52.4 | 49.6 | 57.4 | 61.1 | **49.1** |
| | R | 50.6 | 54.5 | 62.3 | 50.8 | 53.6 | 52.0 | 70.4 | 63.4 | **57.2** |

## 4 EXPERIMENTS

In this section, we describe implementation details of our approach, and provide a quantitative comparison against state-of-the-art methods on standard benchmarks (Straub et al., 2019; Ye et al., 2024). We then present an in-depth analysis along with ablation studies. We also provide quantitative results on ScanNet (Dai et al., 2017) and a qualitative comparison on Mip-NeRF 360 (Barron et al., 2022) in Appendix A.

### 4.1 IMPLEMENTATION DETAILS

**Datasets.** We mainly perform experiments on Replica (Straub et al., 2019) and LERF-Mask (Ye et al., 2024). Following Gaga (Lyu et al., 2024), we select eight indoor scenes from the Replica dataset, with each scene containing 180 training and 180 test images. The LERF-Mask dataset is based on LERF (Kerr et al., 2023), and consists of three scenes: figurines, ramen, and teatime. Each scene provides 6-10 text queries for objects, along with manually annotated masks.

**Training.** We build our method on the official implementation of 3DGS (Kerbl et al., 2023). Following the common practice (Lyu et al., 2024; Ye et al., 2024), we adopt SAM (Kirillov et al., 2023) with a ViT-H (Dosovitskiy et al., 2021) backbone, and augment each Gaussian with a 16-dimensional embedding, that is, $D$ is set to 16. During training, we render both an image and a feature map from a specific view, and jointly optimize a set of 3D Gaussians and individual classifiers using the Adam optimizer (Kingma & Ba, 2015). Specifically, we adopt a learning rate of 2.5e-3 for the embeddings and 5e-4 for the classifiers. For each scene, we set $\tau$, $\lambda_{CE}$, and the total number of training iterations to 10, 0.05, and 30K, respectively. The values of $\delta_d$ and $\delta$ are chosen as in Gaga (Lyu et al., 2024), i.e., $\delta_d = 0.2$ and $\delta = 0.1$, and we set $\delta_d = \delta_s$ for simplicity. All experiments are performed on a NVIDIA RTX A6000 GPU.

**Evaluation.** We follow the same evaluation protocol as in Gaga (Lyu et al., 2024) on Replica (Straub et al., 2019). Specifically, we measure the performance of instance segmentation in terms of mean intersection-over-union (mIoU), precision, and recall. Please refer to Gaga for a detailed description of each metric. We also report PSNR scores between ground-truth and rendered images to evaluate the performance of novel view synthesis. For the LERF-Mask (Ye et al., 2024) dataset, we follow Gau-Group (Ye et al., 2024), adopting Grounding DINO (Liu et al., 2024)

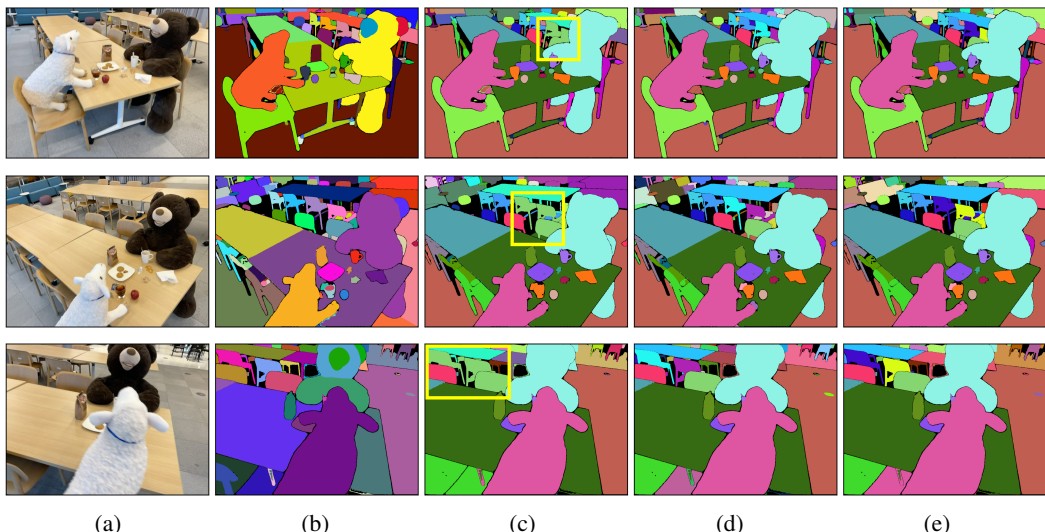

(a)       (b)       (c)       (d)       (e)

Figure 4: Comparison of associated masks on LERF-Mask (Ye et al., 2024). (a) Training images. (b) Inconsistent masks predicted by SAM (Kirillov et al., 2023). (c,d,e) Consistent masks associated by using the depth of each Gaussian $\mathcal{G}_k^{\mathrm{d}}$, the probability of each Gaussian $\mathcal{G}_k^{\mathrm{s}}$, and our method $\mathcal{G}_k^{\mathrm{saga}}$ (from left to right).

to evaluate the performance of open-vocabulary segmentation. We report mIoU and boundary IoU (mBIoU) scores. For all experiments, we report average scores over 3 different runs.

## 4.2 RESULTS

**Replica.** We compare in Table 1 our approach with state-of-the-art methods (Lyu et al., 2024; Ye et al., 2024; Zhu et al., 2025) on Replica (Straub et al., 2019). For a fair comparison, we reproduce all methods using the same set of inconsistent masks predicted by SAM (Kirillov et al., 2023). We also report the results of Unified-Lift trained with consistent masks, denoted by †, obtained from the off-the-shelf video object tracker (Cheng et al., 2023). From this table, we can see that our approach achieves the best performance in terms of mIoU (I), precision (P), and recall (R), demonstrating its effectiveness. In particular, our method produces a precision gain of 6.2% over Gaga, while maintaining a comparable PSNR score.

**LERF-Mask.** Table 2 presents a quantitative comparison of our approach with Gau-Group (Ye et al., 2024), Gaga (Lyu et al., 2024), OmniSeg3D (Ying et al., 2024), and Unified-Lift (Zhu et al., 2025) on LERF-Mask (Ye et al., 2024) in terms of mIoU and mBIoU scores. The results of Gau-Group, Gaga, and OmniSeg3D are borrowed from the paper of Gaga, while we reproduce Unified-Lift. From this table, we have two findings: (1) Unified-Lift trained with inconsistent masks from SAM (Kirillov et al., 2023) performs much worse compared to the variant using the consistent masks, denoted by †. This suggests that Unified-Lift relies heavily on the quality of the associated masks to achieve its best performance. (2) Our approach outperforms all other methods in terms of both mIoU (M) and mBIoU (B), confirming its effectiveness once again.

Table 2: Quantitative comparison with state-of-the-art methods (Lyu et al., 2024; Ye et al., 2024; Zhu et al., 2025; Ying et al., 2024) on LERF-Mask (Ye et al., 2024). I and B represent mIoU and mBIoU, respectively.

| Method | Metric | figurines | ramen | teatime | Avg. |
|---|---|---|---|---|---|
| Gau-Group | I | 69.7 | 77.0 | 71.7 | 72.8 |
| | B | 67.9 | 68.7 | 66.1 | 67.6 |
| Gaga | I | 92.3 | 72.0 | 71.2 | 78.5 |
| | B | 90.8 | 63.3 | 68.4 | 74.2 |
| OmniSeg3D | I | 85.0 | 83.6 | 69.8 | 79.5 |
| | B | 83.7 | 75.5 | 63.8 | 74.3 |
| Unified-Lift | I | 92.0 | 74.0 | 67.8 | 77.9 |
| | B | 90.2 | 68.0 | 64.8 | 74.3 |
| Unified-Lift[†] | I | 87.5 | 76.3 | 77.9 | 80.6 |
| | B | 85.9 | 69.7 | 74.6 | 76.8 |
| Ours | I | 88.2 | 75.0 | 79.4 | **80.9** |
| | B | 86.0 | 69.4 | 75.9 | **77.1** |

## 4.3 DISCUSSION

**Performance of associated masks.** We provide in Table 3 an analysis of each component of our method on Replica (Straub et al., 2019). Specifically, we measure the performance of associated masks on training views. Note that ground-truth masks are used for evaluation only. We can see from the first row that simply using the depth of each Gaussian (Eq. 10) results in a relatively low recall score. On the contrary, we can also see from the second row that using the softmax probabilities (*i.e.*, semantics) of each Gaussian alone (Eq. 15) yields a high recall score of 50.2% at the cost of precision. This suggests that the second criterion produces finer segmentation results, since the precision is inversely proportional to the total number of predicted instances. The last row shows that our approach achieves the best compromise between recall and precision by using both criteria (*i.e.*, depth and semantics of each Gaussian). We also present in Fig. 4 a qualitative comparison on LERF-Mask (Ye et al., 2024). We can see that using the depth of each Gaussian alone fails to separate the three chairs (highlighted by the yellow boxes in Fig. 4(c)), while using the semantics of each Gaussian enables distinguishing them (Fig. 4(d-e)).

Table 3: Quantitative comparison of associated masks on Replica (Straub et al., 2019).

| Association | | | Avg. | |
| Depth | Prob | mIoU | Precision | Recall |
|---|---|---|---|---|
| ✓ | | 42.8 | 16.8 | 46.8 |
| | ✓ | 45.9 | 12.7 | 50.2 |
| ✓ | ✓ | 43.8 | 19.6 | 48.5 |

**Analysis of inference strategies.** To validate the effectiveness of our inference scheme, we compare in Table 4 quantitative results of different strategies on Replica (Straub et al., 2019). Specifically, we explore a training-based strategy, similar to previous methods (Ye et al., 2024; Lyu et al., 2024), where it trains a single classifier with the associated masks. We follow Gaga (Lyu et al., 2024) to train the classifier jointly with the embeddings for 10K iterations, while freezing other attributes of Gaussians. Additionally, we report the results of applying the K-means clustering technique (Arthur & Vassilvitskii, 2006) to a set of prototypes (*i.e.*, $v_t$), obtaining a final classifier for inference without further training. Note that we set the number of clusters to the total number of instance labels within our associated masks. We can see from the first row that using the trained classifier at test time, denoted by Cls, produces unsatisfactory results. We conjecture that this is because training an additional classifier is prone to overfitting to incorrect labels of the associated masks. We can also see from the second row that the strategy adopting the K-Means clustering still underperforms. The last row shows that our strategy using the ensemble of prototypes achieves the best performance, avoiding the overfitting problem.

Table 4: Quantitative results of different inference schemes on Replica (Straub et al., 2019).

| Cls | K-Means | Proto | | Avg. | |
| | | | mIoU | Precision | Recall |
|---|---|---|---|---|---|
| ✓ | | | 47.0 | 41.8 | 53.2 |
| | ✓ | | 51.4 | 48.0 | 55.8 |
| | | ✓ | 51.7 | 49.1 | 57.2 |

**Hyperparameters.** To analyze the effect of varying values of $\lambda_{CE}$ and $\tau$, we provide in Fig. 5 a comparison of mIoU scores on Replica (Straub et al., 2019) and LERF-Mask (Ye et al., 2024). Specifically, we vary $\lambda_{CE}$ with fixing the value of $\tau$ (left), and vice versa (right). We can see from the left that the performance decreases as $\lambda_{CE}$ increases. This is because a large value of $\lambda_{CE}$ prevents 3D Gaussians from reconstructing a given scene, leading to imprecise attributes (*e.g.*, location, color) of each Gaussian. We can also see from the right that extreme values of $\tau$ (either too high or too low) result in suboptimal performance.

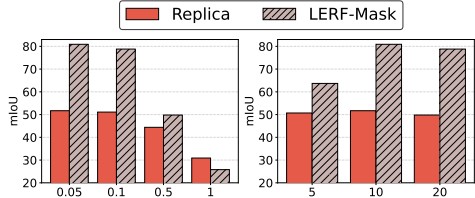

Figure 5: Analysis of different values of $\lambda_{CE}$ (left) and $\tau$ (right) on Replica (Straub et al., 2019) and LERF-Mask (Ye et al., 2024).

## 5 CONCLUSION

We have introduced Proto-SaGa that synthesizes and segments novel views in a 3D scene simultaneously. To this end, we have first designed a simple yet effective training scheme that optimizes a set of 3D Gaussians together with view-specific classifiers. Then, we have proposed a semantic-aware mask association strategy that exploits the learned classifiers to incorporate high-level semantics of each Gaussian during association, improving the consistency of the associated masks. We have also presented a novel inference pipeline using an ensemble of prototypes at test time, reducing the influence of potentially incorrect results from the association process. Finally, we have performed extensive experiments to demonstrate the effectiveness of our approach on standard benchmarks.

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

# Appendix

In the following, we present more results of our method on Replica (Straub et al., 2019), LERF-Mask (Ye et al., 2024), ScanNet (Dai et al., 2017), and Mip-NeRF 360 (Barron et al., 2022) (Sec. A). We also discuss the limitation of our approach (Sec. B)

## A  MORE RESULTS

**Performance of associated masks.**  We show in Table 5 an extended version of Table 3. Specifically, we also report the reproduced results of two baselines: Gau-Group (Ye et al., 2024) and Gaga (Lyu et al., 2024). The first row shows that Gau-Group using DEVA (Cheng et al., 2023) fails to produce consistent masks across training views. The only difference between the second and third rows is that the third one trains all attributes of Gaussians, including embeddings, jointly with individual classifiers. The results indicate that our joint training scheme maintains comparable performance when only the first criterion (*i.e.*, Eq. 10) is used to associate the inconsistent masks.

Table 5: Quantitative comparison of associated masks on Replica (Straub et al., 2019).

| Method | Association | | mIoU | Avg. Precision | Recall |
|---|---|---|---|---|---|
| | Depth | Prob | | | |
| Gau-Group | | | 21.8 | 24.6 | 20.0 |
| Gaga | ✓ | | 43.3 | 16.0 | 47.3 |
| Ours | ✓ | | 42.8 | 16.8 | 46.8 |
| | | ✓ | 45.9 | 12.7 | 50.2 |
| | ✓ | ✓ | 43.8 | 19.6 | 48.5 |

**Hyperparameters.**  Table 6 compares the segmentation performance with varying the values of $\delta_d$ and $\delta_s$. To minimize the need for tuning, we set $\delta_d$ and $\delta_s$ to the same value, *i.e.*, $\delta_d = \delta_s$. We can see from this table that our method achieves the best performance by setting $\delta_d$ and $\delta_s$ to 0.2 on LERF-Mask, while showing the robustness to varying values of $\delta_d$ and $\delta_s$ on Replica.

Table 6: Comparison of segmentation results by varying the values of $\delta_s$ and $\delta_d$ on Replica (Straub et al., 2019) and LERF-Mask (Ye et al., 2024).

| $(\delta_s, \delta_d)$ | LERF-Mask | | Replica | | |
|---|---|---|---|---|---|
| | mIoU | mBIoU | mIoU | Precision | Recall |
| (0.1, 0.1) | 78.1 | 77.8 | 52.0 | 49.3 | 58.0 |
| (0.2, 0.2) | 80.9 | 77.1 | 51.7 | 49.1 | 57.2 |
| (0.3, 0.3) | 74.3 | 74.0 | 51.6 | 49.5 | 57.1 |

**ScanNet.**  We provide in Table 7 quantitative results of our method on ScanNet (Dai et al., 2017). The results of Gau-Group and Gaga are borrowed from the paper of Gaga. We can see that our approach outperforms others in terms of all metrics, confirming its effectiveness.

Table 7: Quantitative comparison with state-of-the-art methods (Lyu et al., 2024; Ye et al., 2024) on ScanNet (Dai et al., 2017). Numbers in bold indicate the best performance, while underscored ones represent the second best.

| Method | mIoU | Avg. Precision | Recall |
|---|---|---|---|
| Gau-Group | 34.2 | 18.7 | 32.6 |
| Gaga | _45.1_ | _22.9_ | _51.0_ |
| Ours | **49.8** | **26.0** | **53.7** |

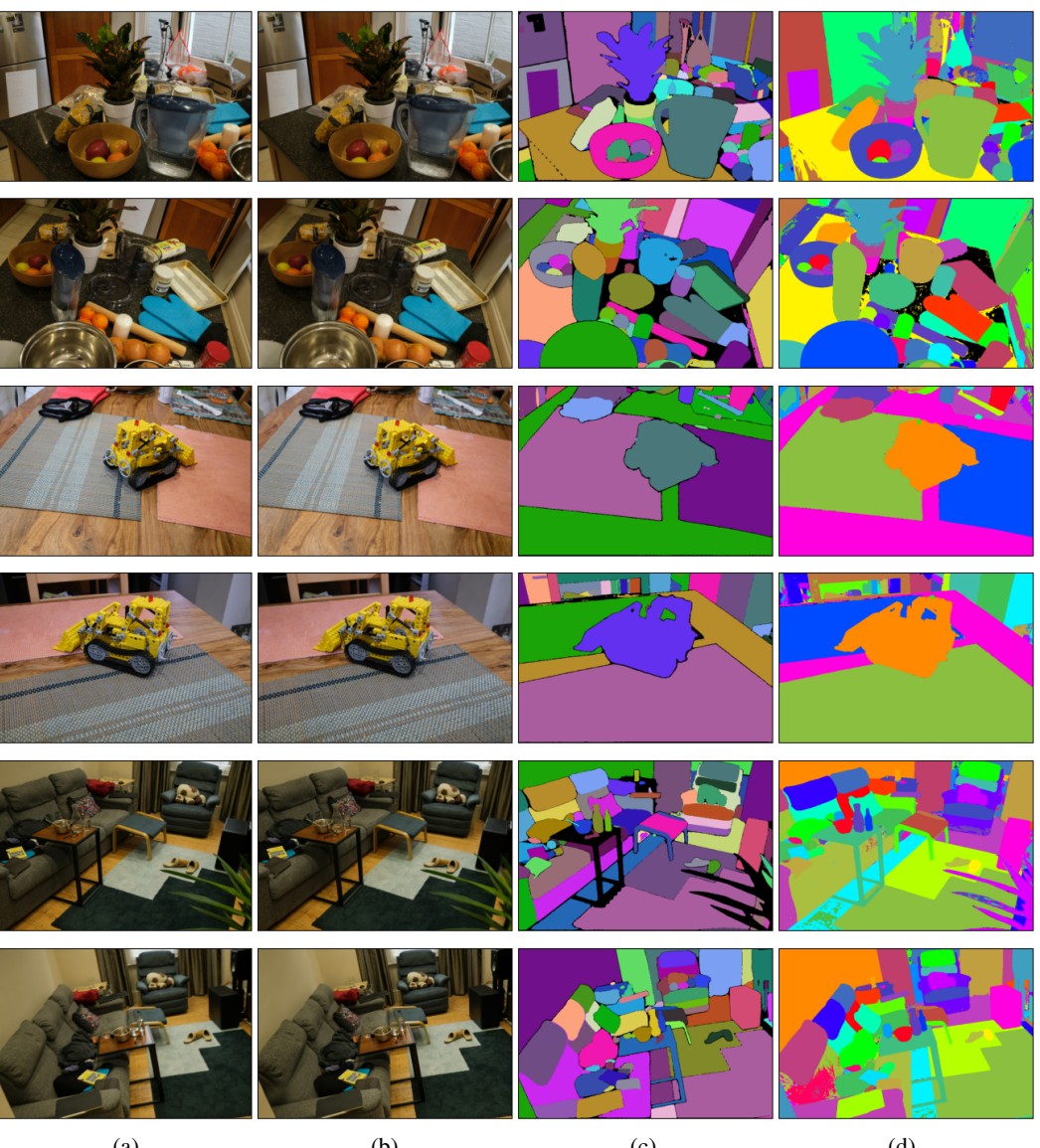

Figure 6: Qualitative results of our method on Mip-NeRF 360 (Barron et al., 2022). (a) Ground-truth images. (b) Rendered images. (c) Inconsistent masks predicted by SAM (Kirillov et al., 2023). (d) Segmentation masks predicted by our method.

**Mip-NeRF 360.** We visualize in Figure 6 results of our approach on Mip-NeRF 360 (Barron et al., 2022). From this figure, we can see that our approach achieves high-quality performance in synthesizing novel view images (Fig. 6(b)), while predicting consistent segmentation masks (Fig. 6(d)).

## B    LIMITATION

Although our approach achieves notable improvements on standard benchmarks (Straub et al., 2019; Ye et al., 2024; Dai et al., 2017), its performance still depends on the quality of inconsistent masks predicted by SAM (Kingma & Ba, 2015) as in current methods (Ye et al., 2024; Lyu et al., 2024; Ying et al., 2024; Zhu et al., 2025). In particular, the inconsistent mask of the first training view is important in that it is used to initialize the memory bank (Eq. 11). A promising direction to address this issue would be designing a method that adaptively selects a specific view whose mask does not suffer from over- and under-segmentation errors.

