# OpenReview forum: "Proto-SaGa: Prototype-based 3D Scene Segmentation with Semantic-aware Gaussian Grouping"
_ICLR.cc/2026/Conference — ICLR 2026 Conference Withdrawn Submission_

### Official Review · Reviewer_x1xX · 2025-10-21

**Soundness:** 2
**Presentation:** 3
**Contribution:** 2
**Rating:** 4
**Confidence:** 4

**Summary:**

This paper works on 3D semantic segmentation. It presents a semantic-aware mask association strategy that exploits both location and high-level semantics of each Gaussian to improve the consistency of the associated masks. Besides, it also proposes an inference scheme that alleviates the influence of possibly incorrect results within the associated masks.

**Strengths:**

1. The proposed medhod uses both location and high-level semantics of each Gaussian, not limited to the location.
2. The inference scheme incroporates prototype-base classifier.
3. The results on standard benchmarks show that the approach achieves a new state of the art.

**Weaknesses:**

1. The first contribution, leverage both location and feature, lacks key insight. The second contribution, add a prototype, is also widely adopted and common. I think the paper lacks key novelty.
2. The reported results of existing works are not consistent with their paper. for instance, Unified-Lift reports 80.9/77.1 on  LERFMask, but this paper reports Unified-Lift result as 80.6/76.8 and the proposed method "80.9/77.1". It shows that the proposed method achieves exactly the same performance with Unified-Lift without improvement.

**Questions:**

Why the reported results of existing works are not consistent with the original paper?

---

### Official Review · Reviewer_uyc2 · 2025-10-30

**Soundness:** 2
**Presentation:** 2
**Contribution:** 2
**Rating:** 4
**Confidence:** 4

**Summary:**

This paper extends the Gaga framework by considering semantic features in the memory bank and introducing an inference algorithm, with effectiveness demonstrated on two datasets. However, the core of the method is based on Gaga, limiting originality, and introducing additional hyperparameters. Ablation studies and comparisons with alternative methods are insufficient to fully validate the approach's advantages.

**Strengths:**

- Based on Gaga, this paper proposes an interesting approach to optimize the semantic features that are added to calculate the memory bank.
- This paper presents a new inference algorithm.
- The effectiveness of the method is verified on two datasets.

**Weaknesses:**

- The entire method appears to be an extended version of Gaga. For instance, the core part still relies on Gaga's memory bank to provide a mask for multi-view consistency, which limits the overall contribution. Additionally, Gaga already suffers from the issues of many hyperparameters, and the incorporation of semantics introduces additional hyperparameters.


- While it is acceptable to add new techniques based on existing baselines, the ablation provided in this paper is insufficient, as it does not demonstrate the effectiveness of the proposed approach compared to existing alternatives. Specifically, while the use of features in the memory bank is shown to improve performance, it remains unclear if the proposed feature learning algorithm is superior to existing methods (e.g., Contrastive learning in [1, 2, 3, 4]). Furthermore, in Table 4, the proposed inference algorithm achieves results similar to the K-means algorithm, and it lacks comparisons to widely used algorithms like HDBSCAN.


- Have the authors conducted comparisons against recent methods like InstanceGaussian[4], given that this baseline also provides instance-level segmentation?

[1] OmniSeg3D: Omniversal 3D Segmentation via Hierarchical Contrastive Learning. CVPR 2024.
[2] Click-Gaussian: Interactive Segmentation to Any 3D Gaussians. ECCV 2024.
[3] Rethinking End-to-End 2D to 3D Scene Segmentation in Gaussian Splatting. CVPR 2025.
[4] InstanceGaussian: Appearance-Semantic Joint Gaussian Representation for 3D Instance-Level Perception. CVPR 2025.

**Questions:**

I am interested in the proposed semantic learning algorithm using individual classifiers and curious about its advantages compared to existing contrastive learning methods. For other questions, please refer to the weaknesses section.

I will consider increasing my score if the above concerns are addressed.

---

### Official Review · Reviewer_Pr27 · 2025-10-31

**Soundness:** 3
**Presentation:** 3
**Contribution:** 3
**Rating:** 6
**Confidence:** 4

**Summary:**

This work proposes a framework for generating 3D-consistent semantic segmentation. To resolve inconsistent segmentation from SAM pseudo labels, the framework adopts the mask association scheme from Gaga and introduces additional guidance using predictions from view-dependent classifiers trained alongside Gaussian splatting. It also proposes a simple prototype-based segmentation scheme to obtain inference labels on novel views. Experimental results show the framework achieves improvements over Gaga and reaches state-of-the-art performance.

**Strengths:**

- The proposed framework improves upon Gaga's mask association scheme and introduces a prototype-based inference scheme. Despite its simplicity, it demonstrates significant improvement over Gaga and achieves state-of-the-art performance.
- Additionally, the proposed mask association scheme is qualitatively evaluated well in Figure 4, and an in-depth analysis is provided in Table 3, which justifies the trade-off between depth and probability criteria.
- Additionally, the effectiveness of inference strategies is well validated in Table 4 (despite some suggestions provided in the Weaknesses and Questions).
- Lastly, the analysis of the hyperparameter provides good insight to the reader regarding the robustness of the proposed framework.

**Weaknesses:**

There are two minor suggestions regarding the current experiments:

- For the Unified-Lift approach, the results using a video tracker for the mask association scheme are reported, which is appreciated. However, since this work improves upon Gaga, it is recommended to also use the mask association scheme from Gaga for Unift-Lift to justify that the proposed association and inference methods are necessary for state-of-the-art performance.
- The inference strategy analysis effectively demonstrates that the proposed framework achieves better results using the prototype approach. To strengthen this analysis, it would be valuable to test the prototype method using masks obtained directly from Gaga. This would isolate the effectiveness of the mask association and provide readers with a clearer understanding of each component's contribution.

**Questions:**

See Weaknesses section.

---

### Official Review · Reviewer_BAcP · 2025-11-01

**Soundness:** 2
**Presentation:** 2
**Contribution:** 2
**Rating:** 2
**Confidence:** 4

**Summary:**

The manuscript introduces a semantic-aware mask association and prototype-based inference framework that jointly leverages geometric and semantic cues within the 3D Gaussian Splatting representation to produce more consistent object segmentation.

**Strengths:**

1. The manuscript proposes a semantic-aware mask association and prototype-based inference framework that jointly leverages spatial and semantic cues of each Gaussian to produce coherent and robust multi-view segmentation.
2. Extensive experiments on standard public benchmarks demonstrate that its method consistently outperforms prior approaches.

**Weaknesses:**

1. Limited technological innovation—it is an incremental extension of existing 3D Gaussian segmentation methods, and the prototype-based classification is a common method in machine learning.
2. F-score should be reported
3. The values in Table 4 are insufficient to determine whether the overfitting issue has been effectively mitigated. More detailed analysis is needed, such as reporting the specific categories and quantities affected by overfitting, along with a clear description of how and to what extent the proposed method improves these cases.
4. The description of the method section needs improvement. Existing techniques should be summarized in the preliminaries subsection, while the proposed contributions should be clearly separated and organized by modules in the method section to highlight the novelty and clarify which parts are original to this work.
5. The results of previous SOTA methods should be reported using the values published in their original papers to ensure a fair and consistent comparison.
6. The results of the proposed method (80.9 / 77.1) are exactly the same as those reported for the previous SOTA, Unified-Lift[1], in its officially published paper, including the decimal precision. Furthermore, the Unified-Lift results are noticeably downscaled in this manuscript, which raises concerns about the credibility of the reported numbers.

[1]Rethinking End-to-End 2D to 3D Scene Segmentation in Gaussian Splatting

**Questions:**

Why does the proposed method achieve exactly the same results on the LERF-Mask dataset (Table 2: 80.9 / 77.1) as the already published Unified-Lift method (Table 1: 80.9 / 77.1)? Is it just a coincidence that two different methods yield identical scores? Moreover, why are the results of SOTA baselines in your manuscript not the officially published results?

---

### Note · Authors · 2025-11-12

I have read and agree with the venue's withdrawal policy on behalf of myself and my co-authors.